# Stability and nature of the volume collapse of ε-Fe$_2$O$_3$ under extreme conditions

J.A. Sans [1], V. Monteseguro[2,3], G. Garbarino[2], M. Gich [4], V. Cerantola[2], V. Cuartero [2,5], M. Monte[2], T. Irifune[6,7], A. Muñoz[8] & C. Popescu [9]

Iron oxides are among the major constituents of the deep Earth's interior. Among them, the epsilon phase of Fe$_2$O$_3$ is one of the less studied polymorphs and there is a lack of information about its structural, electronic and magnetic transformations at extreme conditions. Here we report the precise determination of its equation of state and a deep analysis of the evolution of the polyhedral units under compression, thanks to the agreement between our experiments and *ab-initio* simulations. Our results indicate that this material, with remarkable magnetic properties, is stable at pressures up to 27 GPa. Above 27 GPa, a volume collapse has been observed and ascribed to a change of the local environment of the tetrahedrally coordinated iron towards an octahedral coordination, finding evidence for a different iron oxide polymorph.

[1] Instituto de Diseño para la Fabricación y Producción Automatizada, MALTA Consolider Team, Universitat Politècnica de València, 46022 Valencia, Spain. [2] European Radiation Synchrotron Facility, 38043 Grenoble, France. [3] ICMUV. MALTA Consolider Team, Universitat de València, 46100 Burjassot, Spain. [4] Institut de Ciència de Materials de Barcelona (ICMAB-CSIC), 08193 Bellaterra, Spain. [5] Centro Universitario de la Defensa de Zaragoza, Ctra. Huesca s/n, 50090 Zaragoza, Spain. [6] Ehime University, 2–5 Bunkyo-cho, Matsuyama 790-8577, Japan. [7] Earth-Life Science Institute, Tokyo Institute of Technology, Tokyo 152-8500, Japan. [8] Departamento de Física, Instituto de Materiales y Nanotecnología, MALTA Consolider Team, Universidad de La Laguna, 38207 San Cristóbal de La Laguna, Spain. [9] ALBA-CELLS, 08290 Cerdanyola del Vallés, Barcelona, Spain. Correspondence and requests for materials should be addressed to J.A.S. (email: juasant2@upv.es)

nderstanding the mechanisms behind the formation of the Earth and its differentiation to the present structure has been a subject of intense debate in the last decades[1–3]. The physical and chemical properties of the main constituents of the Earth at specific pressure (P) and temperature (T) conditions played a fundamental role during the differentiation processes, which ultimately lead to the present chemical distribution and layered structure to silicate crust, mantle and metallic core. At present, the chemical composition of the Earth has a few major abundant elements, magnesium, aluminum, silicon and iron combined with hydrogen and/or oxygen or in their metallic state, i.e., iron–nickel alloys in the core. Due to an important presence of oxygen and iron in the Earth's crust and the mantle, binary iron oxides and their derivative compounds can significantly influence the geodynamics of the Earth[4–7]. Consequently, several studies have been performed aiming to investigate the structural behavior of these phases focusing on their stability and their chemical and physical properties at extreme conditions, i.e., high pressure and high temperature (HPHT)[4,8–10].

Studies of mantle rocks show that the oxygen fugacity ($fO_2$) of the upper mantle is relatively high[11] even though the abundance of $Fe^{3+}$ is low due to a negligible incorporation of $Fe^{3+}$ in olivine, the main upper mantle mineral. $Fe^{3+}$ is then readily incorporated into spinel and garnet, as well as into more accessory minerals such as iron oxides hematite ($\alpha$-$Fe_2O_3$) and magnetite ($Fe_3O_4$). In the lower mantle, the favorable coupled substitution of $Al^{3+}$ and $Fe^{3+}$ into (Fe, Mg)$SiO_3$ bridgmanite results in very high bridgmanite $Fe^{3+}/\Sigma Fe$ ratios in equilibrium with metallic Fe[12,13]. Hence, it is possible to conceive the presence of mineral phases with a high content of $Fe^{3+}/\Sigma Fe$ even in the lowermost part of the lower mantle[1]. Indeed, recent studies reported the presence of stable $Fe^{3+}$-rich oxides at conditions relevant for the whole lower mantle[4,9,10,14,15].

Of all iron oxides, the most common is $Fe_2O_3$, with all iron in trivalent state bonded to oxygen forming five different polymorphs[16] at ambient conditions. $\alpha$-$Fe_2O_3$ crystallizes in trigonal structure [space group (s.g.) 167, $R\bar{3}c$, $Z = 6$] named hematite in its mineral form, while the $\beta$-$Fe_2O_3$ belongs to the cubic family [s.g. 206, $Ia\bar{3}$, $Z = 16$][17]. As well as the $\gamma$-$Fe_2O_3$ [s.g. 227, $Fd\bar{3}m$, $Z = 1$] when oxygen vacancies are randomly distributed[18,19] named maghemite in its mineral form. $\varepsilon$-$Fe_2O_3$ belongs to orthorhombic family [s.g. 33, $Pna2_1$, $Z = 8$] and the recently discovered $\zeta$-$Fe_2O_3$ monoclinic structure [s.g. 15, $I2/a$, $Z = 4$][20] was found metastable after a pressure cycle . The alpha phase, thermodynamically stable at ambient conditions and the gamma phases can be found in nature as minerals. Nevertheless, beta and epsilon polymorphs are considered metastable phases synthesized in laboratory[21], only obtained in nanocrystalline form.

The $\varepsilon$-phase contains three octahedral $FeO_6$ units and a $FeO_4$ tetrahedral unit as in the $\gamma$-phase but in a different arrangement (Fig. 1). The strong distortion observed in two of the octahedral units of $\varepsilon$-$Fe_2O_3$ has led some authors[22] to consider their real coordination to be 5 + 1 instead of 6, such that distortion is due to a longer Fe–O bond length respect to its counterparts. These strongly distorted $FeO_6$ octahedral units are similar to those found in some iron-rich silicates[23,24]. Thus, the only known polymorph of $Fe_2O_3$ with Fe in tetrahedral coordination and free from oxygen vacancies is the epsilon phase, whose behavior under pressure is presented here.

The epsilon phase is considered an intermediate product of the transformation between the metastable beta and gamma phases and the thermodynamically stable alpha phase[16,25,26]. In previous studies[27,28], impurities of $\alpha$-$Fe_2O_3$ have been found as a sub-product of the synthesis (up to 10 wt%). However, in the present study, we did not observe any trace of secondary phases within the detection limits of our experimental set-up. This compound

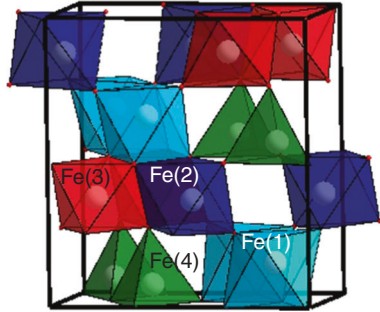

**Fig. 1** Scheme of $\varepsilon$-$Fe_2O_3$ structure. Polyhedral distribution of the $\varepsilon$-$Fe_2O_3$ structure obtained by theoretical simulations. Fe(1)$O_6$ unit is plotted in light blue, Fe(2)$O_6$ unit in dark blue, Fe(3)$O_6$ unit in red, and Fe(4)$O_4$ unit in green

has been the subject of many studies due to its promising magnetic and electric properties[30], such as magnetoelectric coupling, giant coercivity and magnetocapacitance. Besides that, the structural stability of the $\varepsilon$-$Fe_2O_3$ polymorph has exhibited a strong dependence on its particle size. This issue causes to neglect its presence in any geophysical model of the Earth's structure so far. However, the recent findings of this compound in plants[29] in a non-negligible amount (about 10% of the total amount of minerals), the discovery of this material as fundamental part of ancient black-glazed Jian wares[31], as well as the discovery of a isostructural Al-bearing $\varepsilon$-$Fe_2O_3$ mineral (Luogufengite)[32] possibly formed by thermal decomposition of almandine garnet and iron-rich clays in the Earth crust, upper mantle, and transition zone[33], triggered our interest to investigate its stability range at high pressures (HP) and its possible influence in the Earth's geodynamic. Moreover, $\varepsilon$-$Fe_2O_3$ has showed to be highly stable at high temperatures, being the HT polymorph of maghemite[28] above 700 C, and remaining stable up to 1100 C (synthesis conditions) or higher, 1400 C[25]."

This study, combining X-ray diffraction, X-ray absorption spectroscopy (XAS), Mössbauer spectroscopy, and DFT calculations indicate that this compound might be a relevant component of the Earth's interior, and reveals a large structural stability of its outstanding magnetic properties[28,32,34,35]. Moreover, a volume collapse was observed above 27 GPa and was associated to a 1st order structural transition dominated by the change of coordination of Fe(4) from tetrahedral to quasi-octahedral.

## Results

**Angle-dispersive X-ray diffraction under pressure.** In order to assure full hydrostatic conditions, we performed angle-dispersive X-ray diffraction in compressed $\varepsilon$-$Fe_2O_3$ using helium as a pressure transmitting medium. The evolution of the equation of state shows a smooth tendency up to 30 GPa (shown in Fig. 2), whereas in a previous report some anomalies were found at low pressures[37]. Supplementary Figure 2 shows selected XRD patterns at several pressures. The quality of the diffraction patterns allowed performing Rietveld refinements up to 33.7 GPa keeping the large number of free coordinates fixed to the values obtained by ab initio theoretical calculations (Table 1). An example of the quality of the Rietveld refinement is shown in the analysis of the experimental pattern at different pressures exhibited in Supplementary Figure 3, where no traces of additional phases are found. It is worthwhile to notice the stability of the structure up to 27 GPa where a change in the volume occurs (see below). These data were analyzed through a fit to a third-order Birch–Murnaghan equation and allowed us to determine the volume ($V_0 = 424.0(2)$ Angstrom$^3$), bulk modulus ($B_0 = 173(3)$ GPa) and its derivative

($B_0' = 3.8$) at zero pressure. Ab initio theoretical calculations agree nicely with the evolution of the experimental data with a small overestimation of the volume and a comparable bulk

modulus, $B_0 = 168(2)$ GPa with the same derivative ($B_0' = 3.8$), as shown in Fig. 2.

Supplementary Figure 4 illustrates the compression dependence of the lattice parameters, indicating a good agreement with ab initio simulations. The experimental values show a smooth and monotonous decrease under pressure and are fitted to a sublinear EoS in order to determine the compressibility of the axes. Whereas the $b$ and $c$ lattice parameters show a quite similar compressibility ($B_{0b} = 159(3)$ GPa and $B_{0c} = 158(3)$ GPa, respectively) the $a$ crystallographic axis is prominently harder ($B_{0a} = 214(6)$ GPa), which means that the bulk compressibility will be dominated by that of the $b$ and $c$ axis. From these values is straightforward to obtain the bulk modulus of the system ($B_0 = 173(4)$ GPa) through the arithmetical average of the lattice parameters compressibility, which coincides with the value given by the pressure–volume EoS ($B_0 = 173(3)$ GPa). This verification serves to evaluate the goodness of the values reported. On the other hand, the evolution of the ratio between the three lattice parameters is displayed in Supplementary Figure 5 and shows a good agreement between the tendencies given in experimental and theoretically simulated data, validating the structural information extracted from these calculations.

The large number of free coordinates in the Wyckoff sites makes impossible to obtain a reliable result of the interatomic distances by Rietveld refinement of the experimental data (Table 1). Small variations in the determination of those trigger large uncertainties in the bond length of the atoms involved; therefore, further discussions on the structural behavior will rely on theoretical ab initio calculations and the conclusions will be assessed by experimental X-ray absorption spectroscopy and Mossbauer spectroscopy studies. As previously mentioned, ε-$Fe_2O_3$ is formed by three octahedral units and a tetrahedral unit (Fig. 1). The evolution of the polyhedral units with compression is shown in Fig. 3a. The compression of the unit cell is dominated by the compression of the softer polyhedral unit, in this case, the octahedron around Fe(2). This octahedron shows similar bulk modulus as the unit cell volume ($B_{0octFe(2)} = 168.1(15)$ GPa). The octahedron around Fe(1) exhibits a similar compressibility ($B_{0octFe(1)} = 175(3)$ GPa). The octahedral unit around Fe(3) is the hardest, with a $B_{0octFe(3)} = 198(5)$ GPa and exhibits similar compressibility as the $FeO_6$ octahedral unit in hematite compound ($B_0 = 209$ GPa)[36] and that given in the $FeO_6$

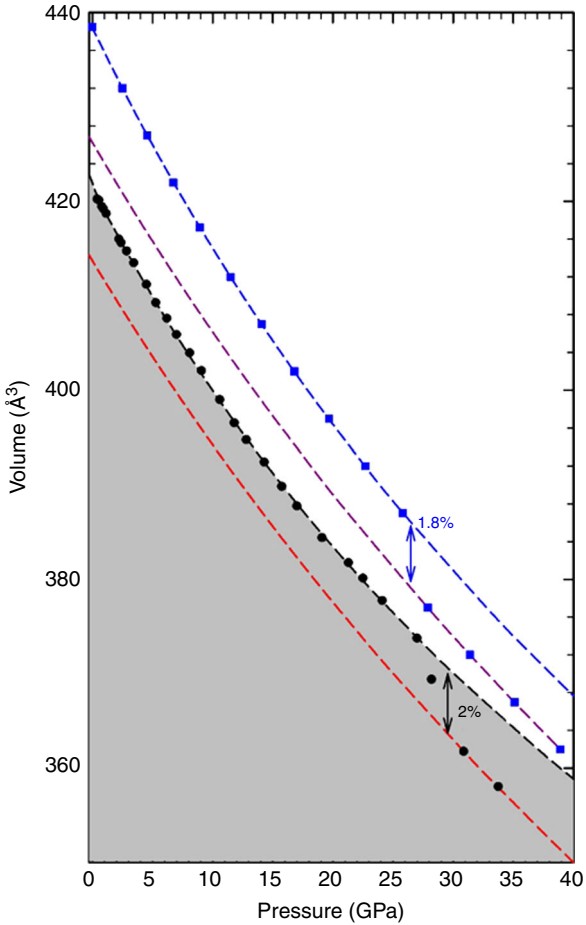

**Fig. 2** Evolution of the volume under compression. Experimental data (black circles), theoretically simulated data (blue squares) and the fits to 3rd order Birch–Murnaghan equation (dashed line)

**Table 1 Crystal data from XRD for the structural phase below and above the volume collapse**

|  | ε-$Fe_2O_3$ (27.0 GPa) | | | ε'-$Fe_2O_3$ (30.9 GPa) | | |
|---|---|---|---|---|---|---|
| Structure obtained by Rietveld refinement | | | | | | |
| s. g. | $Pna2_1$ (No. 33) | | | $Pna2_1$ (No. 33) | | |
| $a$ (Å) | 4.9143(5) | | | 4.8404(8) | | |
| $b$ (Å) | 8.3978(7) | | | 8.2038(10) | | |
| $c$ (Å) | 9.0569(9) | | | 9.1106(15) | | |
| $V$ (Å³) | 373.77(4) | | | 361.78(7) | | |
| $Z$ | 8 | | | 8 | | |
| Fe(1) | 0.8040 | 0.8440 | 0.1460 | 0.8029 | 0.8447 | 0.1453 |
| Fe(2) | 0.8177 | 0.5292 | 0.3625 | 0.8151 | 0.5306 | 0.3635 |
| Fe(3) | 0.8101 | 0.1595 | 0.3695 | 0.8086 | 0.1592 | 0.3691 |
| Fe(4) | 0.7037 | 0.3573 | 0.0642 | 0.7140 | 0.3605 | 0.0666 |
| O(1) | 0.4869 | 0.5163 | $-4.39 \times 10^{-3}$ | 0.4896 | 0.5161 | $-4.23 \times 10^{-3}$ |
| O(2) | 0.5126 | 0.8318 | $-4.91 \times 10^{-3}$ | 0.5127 | 0.8335 | $-4.98 \times 10^{-3}$ |
| O(3) | 0.5477 | 0.1720 | $-1.71 \times 10^{-3}$ | 0.5545 | 0.1757 | $-2.22 \times 10^{-3}$ |
| O(4) | 0.6459 | $2.01 \times 10^{-3}$ | 0.2559 | 0.6408 | $3.06 \times 10^{-3}$ | 0.2542 |
| O(5) | 0.6543 | 0.3375 | 0.2603 | 0.6520 | 0.3378 | 0.2608 |
| O(6) | 0.6447 | 0.6678 | 0.2401 | 0.6426 | 0.6680 | 0.2395 |
| Quality | | | | | | |
| $R$ (%) | 5.2 | | | 7.7 | | |
| $R_{wp}$(%) | 9.5 | | | 13.9 | | |

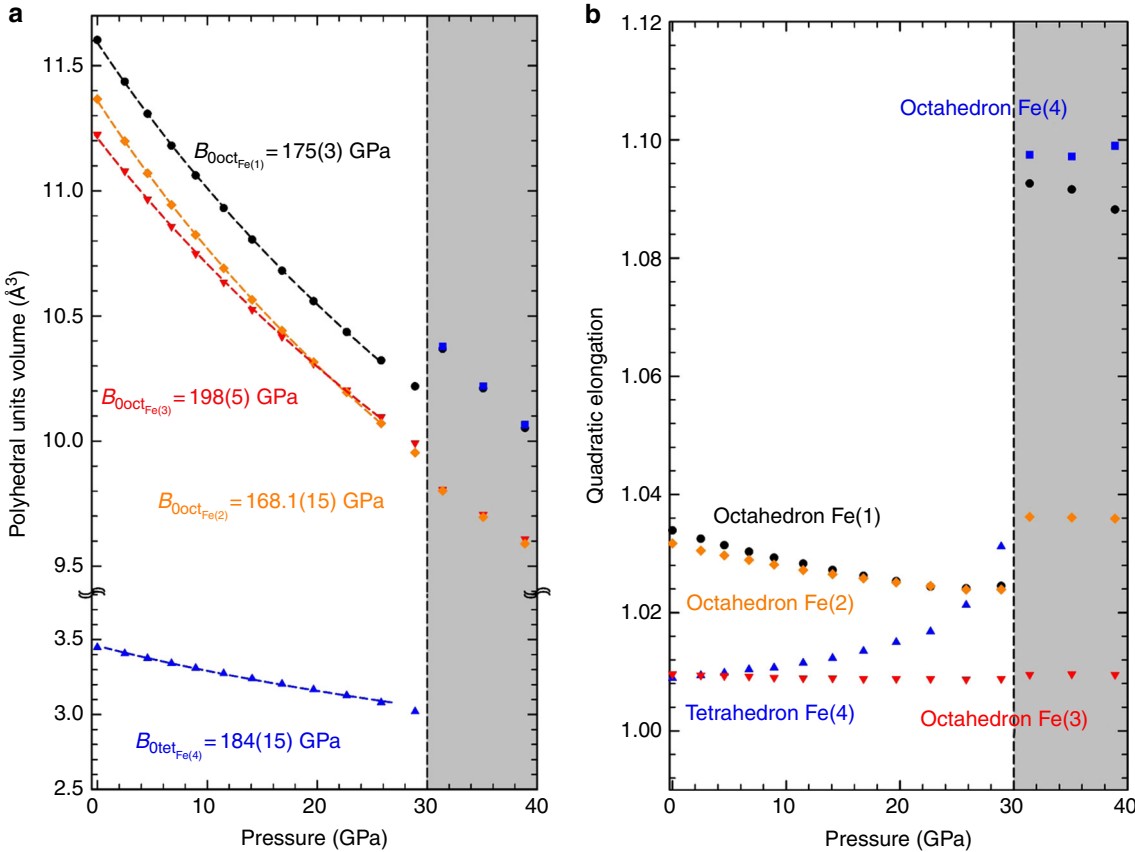

**Fig. 3** Analysis of the volume and the quadratic elongation of the polyhedral units under pressure. Representation of the evolution under compression of theoretically simulated **a** polyhedral unit volume and **b** quadratic elongation of polyhedral units around Fe(1) (black circle), Fe(2) (green diamond), Fe(3) (red triangle down) and Fe(4) (blue triangle up before 40 GPa and blue square after 30 GPa)

octahedron of $Ca_3Fe_2[SiO_4]_3$ ($B_0 = 195(2)$ GPa)[38]. The compression of the tetrahedral unit reveals a bulk modulus of $B_{0tetFe(4)} = 184(15)$ GPa comparable to that of the $Fe(1)O_6$ octahedral unit. In order to better explain the polyhedra deformation we consider the quadratic elongation parameter[39] as shown in Fig. 3b, which is defined by $\lambda = \sum_{i=1}^{n} \frac{(l_i/l_0)^2}{n}$ being $l_0$ the interatomic distance corresponding to an unstrained polyhedral unit, $l_i$ the distances between the cation and the surrounding anions and $n$ the degree of the polyhedral unit, 6 for an octahedron and 4 for a tetrahedron. As expected, the most incompressible octahedral unit (around Fe(3)) keeps its strong regularity along the pressure. The octahedra corresponding to Fe(1) and Fe(2) are distorted at ambient conditions but under compression tend to form a regular octahedral unit, which implies changing from 5+1 to 6 coordination. A special case is the tetrahedron. Even though Fe(4) is almost perfectly centered in the polyhedral unit at ambient conditions, the quadratic elongation indicates that it becomes more irregular with compression. The polyhedral units around Fe(1), Fe(2), and Fe(4) tend to the same value of the quadratic elongation at 25 GPa, before the volume collapse. This quadratic elongation (1.026) corresponds to that observed in the octahedral units of the most stable polymorph of iron oxide, $\alpha$-$Fe_2O_3$. It is noteworthy that the same quadratic elongation has been observed in other iron oxide polymorphs such as the beta-phase[40]. On the other hand, the value of the quadratic elongation of the octahedral unit around Fe(3) is around 1.008 and remains almost unalterable with compression along the stability range of this phase. This parameter is quite similar to

those observed in other polymorphs such as $\gamma$-$Fe_2O_3$ (with $\lambda$ equal to 1.0024 for the octahedron and 1.000 for the tetrahedron)[41], the high pressure monoclinic-$Fe_2O_3$ (with two out of the three octahedral units with a $\lambda$ of 1.0025 and 1.0021, respectively)[42], and $\zeta$-$Fe_2O_3$ (an octahedron with $\lambda$ of 1.0109)[43].

Another parameter used to probe the polyhedra irregularity is the distortion index displayed in Supplementary Figure 6 in which one can distinguish two different behaviors below 25 GPa, where the volume collapse occurs. The oxygen octahedra around Fe(1) and Fe(2) present a decrease in the distortion index with pressure, which differs significantly from the hematite counterpart. This result is supported by the different bulk modulus observed in the octahedral units of both compounds. On the other hand, the octahedron around Fe(3) behaves quite similarly to the $FeO_6$ octahedron in hematite. An increase of the distortion index with pressure and a similar compressibility reveals that the $Fe(3)O_6$ could be considered a common characteristic to different iron oxide polymorphs. Their hardness under compression exhibits as well an extraordinary stability.

The pressure–volume EoS indicates an experimental volume collapse above 27 GPa, where the X-ray diffraction pattern shows no evidence of a structural change but a shift in the position of the Bragg reflections (Supplementary Figure 2). This finding together with the absence of new Bragg peaks leads to the assumption of a collapse of the volume without any change of structure-type. Above 27 GPa, the experimental volume shrinks by 2% which is well reproduced by theoretical ab initio calculations (1.8%) at a slightly smaller pressure (≈25 GPa).

**X-ray absorption spectroscopy under pressure**. Extended X-ray absorption fine structure (EXAFS) under pressure can provide complementary structural information and confirm the results found by ab initio theoretical calculations. The EXAFS signal weighted in $k^2$ and the modulus of its Fourier transform (FT) at pressures below and above the pressure at which the volume collapse occurs are presented in Fig. 4. The EXAFS signals, extracted in a conventional way using the Demeter package[44], display a good quality up to $k = 10\ \text{Å}^{-1}$. The $k^2\chi(k)$ weighted EXAFS signals (Fig. 4a) were Fourier transformed (FT) in the interval $k = 3–10\ \text{Å}^{-1}$ using a Hanning window, as shown in Fig. 4b–d. The structural analysis was performed by using the theoretical phases and amplitudes calculated with FEFF-8 code[45]. Figure 4c, d show the modulus and imaginary part of the Fourier transform (black symbols) and the corresponding best-fitting (red and blue solid lines) including the first and second shells at selected pressures (0 and 31 GPa). The first and second shell analysis was performed in R-space, between 1 and 3 Å$^{-1}$ using Artemis program. The Fe–O and Fe–Fe distances were assumed to follow one single-peak average distribution, assuming that the coordination numbers are fixed to the theoretical values. Therefore, the free fitting parameters were: the deviation from the average distance ($\Delta R$), the Debye–Waller factor ($\sigma^2$) for each of the two Fe–O and Fe–Fe average distance distributions, $S_0^2$ (amplitude reduction factor) and $\Delta E_0$ (edge energy mismatch between theory and experiment). $S_0^2$ was left as a free parameter at ambient conditions and kept fixed to 0.66(6) independently of the pressure. This low amplitude reduction factor is characteristic of iron atoms[46–48]. $\Delta E_0$ was fixed to −0.62 (7) eV below and above the transition, obtained from a first fit at ambient pressure performed with $\Delta E_0$ left free.

The first peak observed in the radial distribution function corresponds to the oxygen environment of iron, uncorrected for the phase shifts, while the second one accounts for the contribution of iron-iron scattering paths. The starting model used for the fitting of the FT signals takes into account the ambient conditions nearest-neighbor Fe–O distances of the two distorted $FeO_6$ octahedra around Fe(1) and Fe(2), a regular $FeO_6$ octahedron around Fe(3) and a regular $FeO_4$ tetrahedron around Fe(4) (Table 2). The best fit, at 31 GPa, was obtained considering two distorted $FeO_6$ octahedra and two regular $FeO_6$ octahedra.

**Synchrotron-based Mössbauer spectroscopy under pressure**. In order to shed more light on the subtle local rearrangement of the new phase, synchrotron-based Mössbauer spectroscopy (SMS) technique was employed providing us with the hyperfine parameters of all crystallographic iron sites upon compression before and after the phase transition (Fig. 5).

The Mössbauer spectra at 293 K and different pressures are complex. The spectrum at 1 bar is in good agreement with those collected in the literature[25,27] at ambient conditions and is characterized by four components with similar relative areas. However, since two of the components are indistinguishable at 1 bar, we follow the same approach described by Popovici and co-workers[27] attributing a unique magnetic sextet (instead of two) for the two irregular octahedral sites Fe(1) and Fe(2) (blue sextet in Fig. 5 top). Hence, from hereafter, we will discuss the crystallographic sites Fe(1) and Fe(2) as a unique site Fe(1,2). The other two components are also sextets, ascribed to the octahedral Fe(3) (red sextet in Fig. 5) and the tetrahedral Fe(4) (green sextet in Fig. 5). Additionally, the gray component in Fig. 5 is the contribution from traces of iron in the confocal Be-lenses of ID18 beamline at the ESRF (see Methodology).

At 1 bar, the Fe(1,2) octahedral site is characterized by hyperfine field (BHF) of 44.5(1) T, the second site corresponding to Fe(3) exhibits a smaller hyperfine field, 39.2(2) T, whereas the tetrahedral site has a BHF unusually low, 25.4(4) T[25]. Note the different values of quadrupole shift (QS) for each site, which arises from the quadrupole interaction with the electric field gradient (EFG), hence expressing a certain degree of distortion, with the Fe(3) site showing the lowest distortion. From the QS of the tetrahedral site (−0.38(10) mm·s$^{-1}$), one could evidence a small distortion, which is not detected by diffraction or reported in previous works. This difference could derive by the difficulty of fitting such complex spectra, which could lead to small variations in the final hyperfine values, but nevertheless consistent with the general results.

Upon compression to 27.5(1) GPa, close to the volume collapse, the spectrum changes completely. First, an extra doublet is observed (orange feature in Fig. 5 center, Table 3). From the CS (0.35(2) mm·s$^{-1}$), we can assign it to a paramagnetic high-spin state of Fe$^{3+}$ in octahedral coordination, whose abundance increases with P[49]. The CS and QS values of this doublet are similar to those found in the literature[50,51] at room conditions, where they were tentatively assigned to the presence of hematite

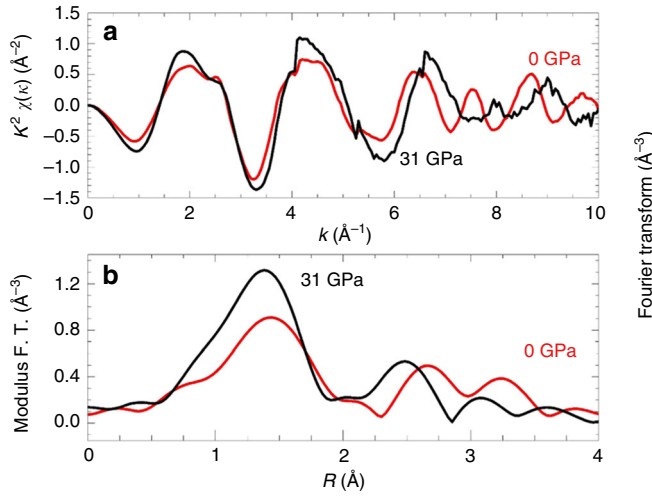

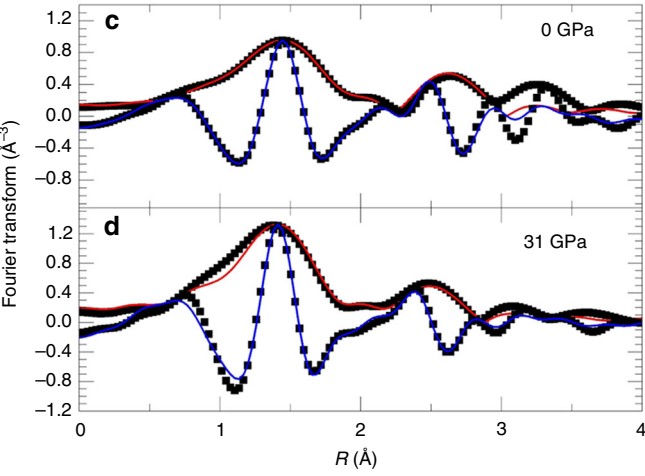

**Fig. 4** Analysis of the EXAFS signals before and after the volume collapse. **a** $k^2\chi(k)$ EXAFS signals and **b** FT at the Fe K-edge at ambient (red line) and 31 GPa (black line). The Fourier transformed $k^2\chi(k)$ EXAFS signals at 0 GPa (**c**) and 31 GPa (**d**) are shown in black squares together with their best fittings of the modulus (red line) and imaginary parts (blue line)

**Table 2 Interatomic Fe–O bond lengths and Debye–Waller factors obtained by EXAFS analysis with uncertainties expressed in brackets. Ab initio theoretically simulated bond lengths are written in italics. The *R*-factor values of the fit are 0.019 (0.013) for 0 GPa (31 GPa)**

| | 0 GPa | | | 31 GPa | |
|---|---|---|---|---|---|
| | Fe–O (Å) | $\sigma^2$(Å$^2$) | | Fe–O (Å) | $\sigma^2$(Å$^2$) |
| Distorted octahedra1 | 2.04(3) *2.08* | 0.008(2) | Distorted octahedra1 | 1.93(1) *2.11* | 0.012(4) |
| Distorted octahedra2 | 1.92(3) *1.96* | 0.008(2) | Regular octahedra1 | 1.87(4) *1.99* | 0.0037(6) |
| Regular octahedra | 1.95(2) *1.94* | 0.003(2) | Regular octahedra2 | 1.89(4) *2.01* | 0.0037(6) |
| Regular tetrahedra | 1.82(1) *1.89* | 0.003(2) | Distorted octahedra2 | 1.94(1) *2.12* | 0.012(4) |

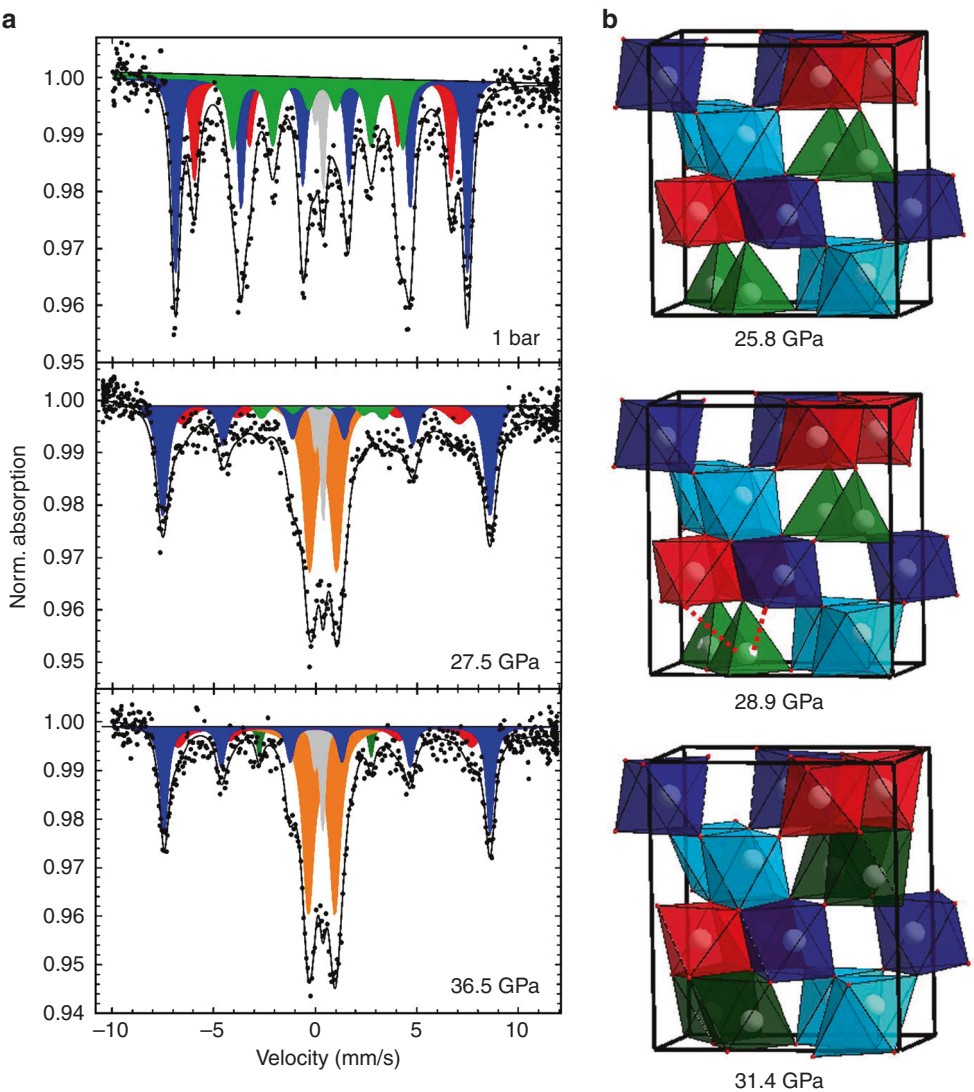

**Fig. 5** Synchrotron-based Mössbauer spectroscopy of ε-Fe$_2$O$_3$ collected before, during and after the volume collapse. **a** Mössbauer spectra of the ε-Fe$_2$O$_3$ below (1 bar), during (27.5 GPa) and above (36.5 GPa) the critical pressure. Blue sextet: Fe(1,2) irregular octahedral sites; red sextet: Fe(3) regular octahedral site; green sextet: Fe$_{tet}$(4) regular tetrahedral site; dark green sextet: Fe$_{oct}$(4) regular octahedral site after transition from tetrahedral geometry; orange doublet: paramagnetic high-spin Fe$^{3+}$ state, caused by the progressive loss of hyperfine field in all iron sites upon compression; gray doublet: contribution from iron in Be-lenses. **b** Structural layout below, during and above the critical pressure

or superparamagnetic (SPM) ε-Fe$_2$O$_3$. In our case, however, the presence of the hematite is excluded by XRD technique and synchrotron Mössbauer spectroscopy, where the hematite, sextet characterized by a large hyperfine magnetic splitting, is not observed. The emergence of the paramagnetic doublet is

accompanied by the near disappearance of the sextet associated to the tetrahedral component Fe$_{tet}$(4), now representing only 7% of the spectrum area and with its hyperfine field dropping to 18.6 (10) T. Regarding the octahedral sites, at 27.5(1) GPa the relative areas of Fe(1,2) and Fe(3) also decrease (Table 3) and present

**Table 3 Hyperfine parameters derived from room temperature SMS spectra of ε-Fe₂O₃. Uncertainties in the hyperfine parameters are indicated in brackets. Component column refers to the color in which each site is displayed in Fig. 5**

| Pressure (GPa) | Fe site | Component | CS[a] mm•s⁻¹ | QS[b] mm•s⁻¹ | Area % | FWHM[c] mm•s⁻¹ | BHF[d] T |
|---|---|---|---|---|---|---|---|
| 1 bar | Fe(window) | Doublet (gray)[e] | 0.15 | 0.41 | 5.4 | 0.27 | — |
| | $Fe^{3+}$(1,2) | Sextet (blue) | 0.37(3) | −0.05(5) | 43.0(21) | 0.46(7) | 39.1(2) |
| | $Fe^{3+}$ (3) | Sextet (red) | 0.39(2) | −0.22(2) | 24.5(21) | 0.40(4) | 44.5(9) |
| | $Fe^{3+}$ (4) | Sextet (green) | 0.21(3) | −0.19(7) | 27.1(27) | 0.56(9) | 25.9(3) |
| 27.5 GPa | Fe(window) | Doublet (gray)[e] | 0.15 | 0.41 | 6.1 | 0.27 | — |
| | | Doublet (orange) | 0.35(2) | 1.33(4) | 32.5(3) | 0.71(6) | — |
| | $Fe^{3+}$(1,2) | Sextet (blue) | 0.32(3) | 0.39(5) | 36(2) | 0.64(6) | 49.9(1) |
| | $Fe^{3+}$ (3) | Sextet (red) | 0.18(14) | 0.1(3) | 18.1(13) | 1.4(3) | 42.6(9) |
| | $Fe^{3+}$ (4) | Sextet (green) | 0.48(17) | −0.3(3) | 7.1(24) | 0.7(4) | 18.6(10) |
| 36.5 GPa | Fe(window) | Doublet (gray)[e] | 0.15 | 0.41 | 6.9 | 0.27 | — |
| | | Doublet (orange) | 0.31(2) | 1.30(3) | 43(3) | 0.64(5) | — |
| | $Fe^{3+}$(1,2) | Sextet (blue) | 0.30(2) | 0.54(5) | 30(2) | 0.44(5) | 49.7(1) |
| | $Fe^{3+}$ (3) | Sextet (red) | 0.21(12) | 0.5(2) | 15.0(12) | 1.0(3) | 44.8(8) |
| | $Fe^{3+}$ (4) | Sextet (dark green) | 0.13(6) | −0.24(13) | 6(2) | 0.26(19) | 17.1(5) |

[a]CS: center shift relative to α-Fe
[b]QS: quadrupole splitting
[c]FWHM: full width at half maximum including the source linewidth
[d]BHF: magnetic hyperfine field
[e]Doublet contribution: iron in confocal Be-lenses, ID18[69]

significantly broadened linewidths compared to the 1 bar spectrum with a 56% increase for Fe(1,2) and 250% increase for Fe(3). In contrast to the behavior of the Fe$_{tet}$(4), the hyperfine fields of the octahedral sites Fe(1,2) and Fe(3) increase with the compression to 49.9(1) and 42.6(9) T, respectively. Furthermore, the octahedral sites Fe(1,2) and Fe(3) at 27.5(1) GPa are characterized by lower CS values respect to the spectrum at ambient P (Table 3), which indicates a higher density of s electrons around the ⁵⁷Fe nuclei[52]. The QS values go from negative to positive (Table 3), which might indicate a change in the direction of the magnetic moment, but in order to test this hypothesis, further experiments will have to be performed under the application of opposite external magnetic fields.

Upon increasing the pressure to 36.5(1) GPa, the contribution of all the sextets to the spectrum continues to decrease with a corresponding increase of the doublet area (Fig. 5 bottom, Table 3). In particular, at this pressure, the relative area of the blue and red sextets associated to octahedral sites has decreased ~36% with respect to their value at 1 bar and in the case of the tetrahedral Fe(4), this diminution is 75%. Note that at the highest pressure, the linewidth of the blue sextet is comparable to its value at 1 bar. In contrast, for the red sextet associated to Fe(3) the linewidth is more than twice its value at 1 bar, even though it has significantly decreased compared to its value near the volume collapse.

## Discussion

A volume collapse similar to that exhibited by ε-Fe₂O₃ at about 27 GPa, has already been studied in other iron-based compounds such as siderite[9,53], ferropericlase[54], (Mg,Fe)SiO₃ perovskite[55], or RFeO₃ perovskites[58] and has been associated to a high-spin to low-spin (HS-LS) transition of iron. Similar behavior has been observed in hematite, where the origin of this HS-LS transition has been the subject of multiple discussions[56,57]. In the present case, the nature of the volume collapse in ε-Fe₂O₃ is properly explained by the local structural change around Fe atoms leading to a new phase of epsilon Fe₂O₃. According to the local structural analysis, the evolution of the theoretical quadratic elongation of the tetrahedral unit shows an increasing tendency of the Fe(4) atom to shift out of the central position (Fig. 5). This effect is

emphasized above 28.9 GPa, where Fe(4) atoms suffer an abrupt change in their quadratic elongation. This feature indicates that Fe(4) atoms tend to shift away from the central position within the tetrahedron, establishing an octahedral coordination by interacting with an oxygen atom belonging to the octahedra formed by Fe(1) atoms above 28.9 GPa. Therefore, Fe(1)O₆ and Fe(4)O₆ octahedral units become quite similar in volume and in quadratic elongation while corresponding to the same layer perpendicular to the c-axis.

These results are corroborated by EXAFS and Mössbauer spectroscopy. Regarding the Debye–Waller factor in the EXAFS fits, below 27 GPa, it is interesting to highlight that values around 0.003 correspond to regular polyhedral units and larger values, like 0.008, imply distorted polyhedral (Table 2). Thus, the comparison of average Fe–O bond length and the Debye–Waller factor with ab initio theoretical calculations allowed us to assign the results to the presence of two distorted octahedral units (Fe(1)O₆ and Fe(2)O₆), a regular octahedron (Fe(3)O₆) and a regular tetrahedron (Fe(4)O₆) at ambient conditions. Above 27 GPa, one can observe two regular and two distorted polyhedral units, according to the Debye–Waller factor, in agreement with the theoretically predicted structure. Although the regular Fe(3)O₆ octahedron remains unalterable, the distorted octahedral unit corresponding to a Fe(1)O₆ increases the distortion, as observed by the quadratic elongation analysis. The Fe(2)O₆ octahedron becomes regular above the critical pressure and is similar to the regular Fe(3)O₆, with which shares the same layers perpendicular to the c-axis. Nevertheless, the most striking result is the evolution of the tetrahedron Fe(4)O₄. It is possible to observe how the average Fe–O distance, as well as the Debye–Waller factor (0.012), increases with pressure towards values similar to Fe(1)O₆ distorted octahedron. As far as the local structure is concerned, these parameters indicate that the Fe(4) tetrahedral unit observed at low pressures increases its coordination towards a distorted octahedron above 27 GPa, since the correlation parameter using an octahedron instead of a tetrahedron is much smaller, confirming the structure obtained by theoretical simulations. Paying special attention to the arrangement of octahedral units after the volume collapse, the application of pressure gives rise to the arrangement of alternate layers of two blocks perpendicular to the c-axis. Each block is formed by two different octahedral units

quite similar among them; on one hand, irregular Fe(1)O$_6$ and Fe (4)O$_6$ and on the other hand, regular Fe(2)O$_6$ and Fe(3)O$_6$ polyhedral units. The change in the coordination observed in the Fe(4) could indicate a phase transition. Although the crystallographic symmetries and multiplicities are equal to the low pressure phase (Table 1), a change in the electronic topology of the iron cations is enough to consider this new structure as the ε' phase.

The question about the origin of the structural change remains open. Our results unambiguously indicate the change of coordination of Fe(4) from tetrahedral to an octahedral unit; however, this feature does not exclude the possibility of HS-LS electronic transition. In order to solve this, the interpretation of the results obtained by Mössbauer spectroscopy and the theoretical calculations become relevant. The interpretation of the hyperfine parameters of the Fe$_{tet}$(4) site at 27.5(1) GPa is complex. However, one can notice the drastic increase of the CS from 0.10(6) mm·s$^{-1}$ to 0.48(17) mm·s$^{-1}$, which reflects the transition from tetrahedral to an octahedral site, the site becoming slightly larger and consequently diminishing the s electron density at the Fe-nuclei. Above the critical pressure, the octahedral sites Fe(1,2) and Fe(3) have CS consistent with the previous values at 27.5(1) GPa (Table 3), but now present much higher QS, respectively of 0.54(5) and 0.49(23) mm·s$^{-1}$, which could describe slightly more distorted local environments than at lower pressure. Indeed, it is common to observe the increase of the QS at high pressures[52], since compression can affect the distribution of the EFG, especially if the hydrostatic conditions are not nicely preserved.

Two scenarios were explored to explain the origin of the paramagnetic doublet at HPs. On one hand, the presence of small enough nanoparticles to exhibit SPM properties, which has been dismissed by the calculation of the critical SPM volume that estimates a diameter of <2 nm to exhibit these properties. Thus, the reduction of the nanoparticle size by compression cannot explain the presence of this doublet at HPs. On the other hand, an ongoing pressure-induced amorphisation (PIA) in some of the nanoparticles, which is not supported since the appearance of the doublet seems to be mostly related to the decrease of the Fe(4) sextet area, i.e., would only affect to the tetrahedrally coordinated iron. Moreover, the peaks of this doublet are not broad enough and are perfectly symmetric to consider an amorphous origin. Similar behavior has been observed in previous investigations of other iron-based compounds at extreme conditions[58], possibly indicating in the present case a weakening of the ferromagnetic interactions, mostly affecting the Fe(4) sublattice, associated to the phase transition and the change in the coordination environment of the Fe(4) site. In fact, in the case of hematite, the quasi-isostructural transition at 50 GPa associated to a volume collapse is also accompanied by the gradual establishment of a metallic paramagnetic state upon increasing the pressure up to 82 GPa[59]. Additionally, this paramagnetic doublet can also be associated to a reduction of the magnetic anisotropy with the increase of pressure due to a decrease of the crystalline anisotropy.

In the case of the volume collapse of ε-Fe$_2$O$_3$, the observed inversion of the sign of the QS of Fe(1,2) and Fe(3) could indicate a change in the magnetic structure. Indeed, this oxide presents strong spin-lattice coupling and a strong dependence of the unit cell volume on the magnetic ordering[28]. The residual sextet associated to Fe(4) at 36 GPa has a very low hyperfine field, 17.1 (5) T, a CS of 0.13(6) mm·s$^{-1}$, consistent within error with the CS of the Fe(3) site, and a QS of −0.24(13) mm·s$^{-1}$, which reflect a slightly less distorted local environment than at 27.5(1) GPa. According to these results, the weak hyperfine field, the low CS and the high QS of the doublet may suggest that Fe$_{oct}$(4) could be in an intermediate spin (IS) state[60]. The intermediate spin would

explain very well the strong collapse of the hyperfine magnetic field, reducing the spin quantum number of the system in the Fe (4) site from S = 5/2 to S = 3/2, hence weakening but not destroying the magnetic interactions, which would be the case at S = 0. Moreover, since we observe the collapse of the magnetic field in the Fe(4) site before its volume collapse, it is possible to argue that the change in the coordination environment from 4 to 5 + 1 is driven by the electronic transition from HS to IS. This kind of spin crossover from HS to IS has been observed in compounds[61] with pentacoordinated Fe$^{3+}$ such as [Fe$^{3+}$(3 L·)$_2${P (OPh)$_3$}]$^+$ which is quite close to the coordination 5+1 observed around the Fe(4). Moreover, the analysis of the behavior of the theoretically simulated magnetic moments of each iron in the ε-Fe$_2$O$_3$ exhibits values higher than 4 μ$_B$/Fe atom before and after the volume collapse, which are incompatible with a LS configuration but leave the path open to consider an IS state.

The structural stability of the ε-Fe$_2$O$_3$ compound beyond 23 GPa (transition pressure between the upper and the lower mantle), might indicate that the presence of this mineral with paramount magnetic properties has been underestimated in the Earth's interior. This assumption is supported by the recent discovery of nanominerals of ε-Fe$_2$O$_3$ in basaltic rocks[62]. In order to explore its existence under these conditions, we must include the temperature parameter in the theoretical calculations. The structure becomes more energetically favored with the increase of the temperature, comparing the theoretical evolution of the Gibbs free energy vs volume curve at several temperatures (Supplementary Figure 7). The theoretical simulation of the phonon dispersion curve at 25 GPa and 1800 K (Supplementary Figure 8) reveals that the structure is dynamically stable at these conditions by the absence of imaginary or soft modes.

In summary, the analysis of the structure of ε-Fe$_2$O$_3$ at high pressures allowed us to characterize its behavior under compression. ε-Fe$_2$O$_3$ has shown an experimental structural stability up to 27 GPa at ambient temperature and a predicted dynamical stability at 25 GPa and 1800 K revealing its potential presence in the Earth's interior. An anomalous behavior is observed above 27 GPa, corresponding to a volume collapse without any change on the crystallographic planes nor on the symmetry of the material. Ab initio theoretical calculations and combined experimental X-ray diffraction, X-ray absorption and Mössbauer spectroscopy techniques allowed us to conclude that this volume collapse is driven by subtle local structural changes around the Fe atoms and specifically around Fe(4). The characteristic Fe(4)O$_4$ regular tetrahedron of the ε-Fe$_2$O$_3$ compound transforms into a strongly distorted octahedral unit above 27 GPa, which may be related to a HS-IS crossover of the iron in Fe(4) crystallographic site according to the results with X-ray spectroscopic techniques and in good agreement with theoretical calculations. This spin crossover is coherent with iron in 5 + 1 coordination. This is reliable and robust evidence of a 1st order phase transition to a different Fe$_2$O$_3$ polymorph (ε'-phase).

## Methods

**Sample preparation.** ε-Fe$_2$O$_3$ nanoparticles were prepared according to the three-step approach reported in detail in ref. [28]. In the first step silica gels containing iron were obtained from an hydroethanolic sol of tetraethyl orthosilicate (TEOS) of molar composition TEOS:Ethanol:water = 1:6:6 with dissolved iron nitrate non-ahydrate. In a second step the gels were dried and thermally treated at high temperature to obtain a SiO$_2$/ε-Fe$_2$O$_3$ composite. Finally, the SiO$_2$ matrix was removed by etching in hot NaOH. To prepare ~2.3 g of ε-Fe$_2$O$_3$, 5.0 ml of milliQ water and 31.4 ml of absolute ethanol (Panreac) were stirred in a 100 ml beaker, to which 11.7 g of iron nitrate (Aldrich) were dissolved before adding dropwise 20 ml of TEOS (Aldrich). The gels were dried and treated in air atmosphere at 1100 °C for 3 h. The resulting material consisted of single crystalline ε-Fe$_2$O$_3$ nanoparticles embedded in a silica matrix with an average diameter of around 17 nm as observed by transmission electron microscopy (see Supplementary Figure 1). The silica was etched overnight in a 12 M NaOH aqueous solution at 80 °C and the rinsed with

water and re-precipitated by centrifugation several times. The resulting size distribution is shown in Supplementary Figure 1.

**High pressure measurements**. In all the experimental procedures, we use a membrane-type diamond anvil cell consisting in two diamonds faced one to each other. Between them, we use a gasket with a hole in the middle, which defines the pressure cavity where the sample, the pressure calibrant and the pressure transmitting medium are placed. The pressure transmitting medium (in our case helium in the three experiments) assures the application of the pressure occurs hydrostatically. The pressure inside the cavity is obtained by the measurement of a calibrant, in most of the experiments we took advantage of the evolution under pressure of the fluorescence signal of ruby chips inserted into the pressure cavity. The pressure is controlled by the membrane of the diamond anvil cell which is inflated applying force to the mobile anvil whereas the other diamond is fixed and the pressure is monitored by the calibrant.

**X-ray diffraction**. We have performed synchrotron-based X-ray diffraction experiments under compression using helium as PTM at ID27 beamline at the European Synchrotron Radiation Facility (ESRF). The beam was monochromatic with a wavelength of 0.3738 Å and focused to a beam size down to $3 \times 3$ $\mu m^2$ using Kirkpatrick–Baez mirrors. Pressure was determined simultaneously by the luminescence of a ruby chip[63] and the equation of state (EOS) of metallic $Cu$[64] in a cavity drilled in a rhenium gasket. Above 2 GPa, a small rhenium shard seems to be in the pressure cavity giving rise to a weak diffraction signal. Cu is placed beside the sample in order to get a clean sample diffraction pattern. X-ray diffraction images were integrated with Fit2d software[65] and the X-ray diffraction patterns interpreted using the GSAS-EXPGUI package[66] and VESTA software[67].

**X-ray absorption**. We have performed a synchrotron-based micro X-ray absorption spectroscopy experiment under pressure in order to describe the local structure around Fe atoms of $\varepsilon$-$Fe_2O_3$. The experiment has been carried out at BM23 beamline of the ESRF equipped with a double crystal Si (111) monochromator and Kirkpatrick–Baez mirrors to focus the monochromatic X-ray beam to $5 \times 5$ $\mu m^2$, with a Pt coating and set at an angle of 6.5 mrad to reject high order harmonics. Such experiment was performed in transmission mode at Fe K-edge (7.21 KeV) until 35 GPa using a membrane pressure cell with 300 $\mu m$ of cullet nanodiamonds, one of them being partially perforated, to avoid Bragg peaks. The PTM was methanol–ethanol and the luminescence of a ruby was used to measure the pressure in the cell.

**Synchrotron Mössbauer source spectroscopy**. Energy-domain SMS measurements were conducted at the Nuclear Resonance beamline[68] ID18 at the European Synchrotron Radiation Facility (ESRF), Grenoble, during operation in multibunch (7/8 + 1) mode. The SMS is based on a nuclear resonant monochromator employing pure nuclear reflections of an iron borate ($^{57}FeBO_3$) single crystal[69]. The source provides $^{57}Fe$ resonant radiation at 14.4 keV within a bandwidth of 6 neV which is tunable in energy over a range of $\pm 0.6$ $\mu eV$[69].

The X-ray beam emitted by the SMS was focused to a 16-vertical $\times$ 15-horizontal $\mu m^2$ spot size at the full width half maximum (FWHM). Before and after each sample measurement, the SMS linewidth was determined using a $K_2Mg^{57}Fe$ $(CN)_6$ reference single line absorber. The velocity scale ($\pm 12$ mm·s$^{-1}$) was calibrated relative to a 25-$\mu m$-thick natural $\alpha$-Fe foil.

The small cross section, high brilliance and fully resonant and polarized nature of the beam allowed for rapid spectra collection (~6 h per spectrum). Note that $\varepsilon$-$Fe_2O_3$ nanoparticles contained only natural abundance of $^{57}Fe$-atoms, which consist of ~2% of $^{57}Fe$ afu. Due to the scarce amount $^{57}Fe$-resonant atoms, we utilized DACs with the complete volume of the gasket's hole.

The spectra were fitted with a full transmission integral and pseudo-Voigt line shape using the software package MossA[70]. The single line spectra were fitted with a normalized Lorentzian-squared source line shape. A linear function was applied to model the background. In this way we were able to obtain an accurate determination of iron distribution in the different sites of the investigated sample. Note that in order to obtained the maximum amount of photons/s we used the confocal Be-lenses installed at the ESRF beamline ID18. Be-lenses always bear traces of iron that result in the presence of a doublet (gray component in Fig. 5), which is easily distinguished and characterized in all spectra.

**Ab initio total-energy calculations**. Ab initio total-energy calculations have been performed within the framework of density functional theory[71]. The VASP package was used to carry out calculations using the pseudopotential method and the projector augmented-wave scheme (PAW)[72,73].

The calculations were carried out with a unit cell containing 40 atoms. For iron, 14 valence electrons were used ($3p^6 3d^6 4s^2$) whereas 6 valence electrons ($2s^2 2p^4$) were used for oxygen. Highly converged results were achieved by extending the set of plane waves up to a kinetic energy cutoff of 550 eV. In order to provide a reliable description of the effects of electronic correlation the calculations were performed using the GGA + U formalism[74] with the Dudarev's approach. The effective on site Coulomb and exchange parameters were set to $U = 5$ eV and $J = 1$ eV, yielding

reliable results for the magnetic moments and the cell parameters as compared to experiment.

A dense Monkhorst-Pack grid of k-special points was used to perform integrations along the Brillouin zone (BZ) in order to obtain very well converged energies and forces. At each selected volume, the structure was fully relaxed to its equilibrium configuration through the calculation of the forces on atoms and the stress tensor. It should be noted that, within the DFT formalism, the theoretical pressure, $P(V)$, can be determined at the same time as the total energy, $E(V)$, but independently since $P$ (like other derivatives of the energy) can be obtained from the calculated stress[75]. In the relaxed configurations, the forces on the atoms are <0.006 eV Å$^{-1}$ and the deviation of the stress tensor from a diagonal hydrostatic form is <0.1 GPa.

We treat thermal effects within the quasi-harmonic (QHA) approximation[76] by including the vibrational contribution to the Gibbs free energy at constant pressure.

## Data availability
The data sets generated during and/or analysed during the current study are available from the corresponding author on reasonable request.

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

## Acknowledgements

We are thankful for the financial support received from the Spanish Ministry of Economy and Competitiveness for the Ramon y Cajal Fellowship to J.A.S. (RYC-2015-17482), for Juan de la Cierva Fellowship to V.M. (FJCI-2016-27921), and the national projects FIS2017-83295-P, MAT2016-75586-C4-1/2/3-P, MAT2015-64442-R, and the "Severo Ochoa" Programme for Centres of Excellence in R&D (SEV- 2015-0496) co-funded by ERDF of European Union. The Generalitat de Catalunya is also acknowledged for financial support (project 2017SGR765). We acknowledge the ESRF and ALBA synchrotrons for providing beamtime. We also thank D. Santamaría-Pérez and J. Ibañez for fruitful discussions.

## Author contributions

G.G. and C.P. performed the X-ray diffraction measurements under compression at ESRF and ALBA synchrotron that were analyzed by J.A.S. The XAS measurements and analysis were carried out by V.M., V.Cu., and M.M. at ESRF with nanodiamonds provided by T.I. SMS measurements under pressure and analysis were done by V.Ce. The ab initio calculations were done by V.M. and A.M. The sample was provided by M.G. The interpretation of the results and the writing of the manuscript were done by J.A.S., V.M., M.G., V.Ce., V.Cu. and C.P.

## Additional information

**Competing interests:** The authors declare no competing interests.

