## [Peer Review File · Nature Communications]

Reviewers' Comments:

Reviewer #1:

Remarks to the Author:

The manuscript in review by Sans et al. focuses on studying compression behavior of less-common, orthorhombic phase in the Fe₂O₃ system known as epsilon-Fe₂O₃. The existence of the epsilon phase and details of its crystal structure have been known for some time. The epsilon phase has been attracting significant attention because of its interesting magnetic properties, including giant coercive field, magnetoresistance, and millimeter wave ferromagnetic resonance, and is an object of current active investigation in the solid state materials field. To my knowledge and understanding, epsilon Fe₂O₃ is not a phase of significant geologic or geophysical relevance.

The manuscript describes results of three separate in situ high-pressure experiments at ambient temperature with epsilon Fe₂O₃ samples: powder X-ray diffraction, Fe ka-edge EXAFS and synchrotron Mössbauer, over a range of pressure from atmospheric up to about 40 GPa. The experiments seem well designed and executed, and the results are sound and consistent, between the different experiment techniques.

The results indicate continues compressional behavior up to about 25 GPa, followed by a small volume collapse of 2%, which is interpreted in terms of change in magnetic ordering.

For structural refinements the authors use a starting model, which was reported by [22] Kelm and Mader (2005). I believe this model was later debated and a minor revision (in the same space group) was proposed by Gich et al. (2006) Chemistry of Materials, 18, 3889.

I found a report of similar high-pressure experiments on epsilon-Fe₂O₃ (powder XRD and synchrotron Mössbauer), but limited to 12 GPa, published in 2013 DESY annual reports by Brazda et al., but I could not find a corresponding journal publication. Perhaps discussing these results to some extent would be useful (there was an anomaly of compressional behavior at lower pressure, which is not seen in the experiments by Sans et al.).

In my opinion this is sound work, and worthwhile piece of knowledge revealed, but I do not see enough significant broad appeal to justify publication in Nature Communications in the current form. Any kind of more specialized journal would be better suited. I think the authors made a mistake in overplaying the geological context. This phase is of fairly significant interest in solid state materials science, and I would strongly recommend to focus on these aspects. Otherwise the paper is very clean and well-written.

The geological implications are minor at best. These phases (epsilon and epsilon') are not expected to be present in mantle rocks at appreciable quantity. They are almost for certain only thermodynamically metastable. The study is constrained to ambient temperature, which is not the temperature at which rocks exist in the mantle.

In summary, I can't recommend publication in Nature Communications, but would recommend to refocus the paper on materials science aspects and resubmit to a specialized journal.

Reviewer #2:

Remarks to the Author:

In this work, the authors report formation of a new iron(III) oxide polymorph from epsilon-Fe₂O₃ upon exposing it to an elevated pressure. Above a threshold value of the pressure, the structural transformation occurs involving all four cation sites in the crystal structure of epsilon-Fe₂O₃. More importantly, the initially tetrahedral site dramatically changes its environment upon pressure increase adopting features of octahedral coordination. The experimental results were supported by

the theoretical calculations. The authors show the significance of these results achieved; this study can generate and stimulate an eminent interest in the field of paleomagnetism and magnetotellurism.

In general, the work is well organized and well written; all the experimental data acquired are discussed in the logical and sophisticated way and are well supported by the results of the theory used. The Reviewer finds this work very important for identifying new iron oxide phases that can be stable under the conditions in the Earth's crust and that can be a genesis to other iron oxide phases and/or solid solutions. When going through the manuscript, the Reviewer found some minor things and issues that should be cleared up when revising the work:

1. The authors explain the changes in the values of the quadrupole splitting upon pressure increase by changes in the direction of the magnetic moment at the specific sites. It is generally accepted that at room temperature, epsilon-Fe₂O₃ behaves as a collinear ferrimagnet and a model of two-supersublattice ferrimagnet is used to explain its magnetic features at room temperature. In this model, A- and C-sublattice magnetizations from distorted octahedral FeA and regular octahedral FeC sites are parallel to each other and form one superlattice and B- and D-sublattice magnetizations from distorted octahedral FeB and tetrahedral FeD sites form another supersublattice. From the Mössbauer data, it can be inferred that this model is probably not valid for this new phase and the sublattice magnetizations may adopt different orientation with respect to the crystallographic axes of epsilon-Fe₂O₃. It is possible, at this stage, to speculate about the orientation of sublattice magnetizations at elevated pressures for individual sites in the crystal structure of epsilon-Fe₂O₃?

2. Two possible explanations have been suggested for appearance of a doublet upon raising the pressure. The proposed idea of a transition involving emergence of new electronic and magnetic state seems relevant to the Reviewer. However, as the sextets are still present, it implies that only a portion of the epsilon-Fe₂O₃ particles undergo such transition. As the system shows the size distribution (as demonstrated in Figure S1), can this transition be associated with some critical size which is pressure dependent? In other words, as the pressure increases, more and more particles undergo such transition. This could explain the decrease in the spectral areas of sextets and increase in the spectral area of the doublet. Alternatively, this transition can happen at the surface layers and then proceed to the core of the nanoparticles as the pressure gradually increases. Can the authors comment more on this issue?

3. Figure 5a: The fitting seems to be not perfect as the fitted middle and inner lines of some spectral components do not match the experimental data. Is it due to the experimental error or the error of the fitting?

4. If the pressure is released, is the crystal structure of epsilon-Fe₂O₃ fully restored? Are there any deviations in the crystal structure due to pressure treatment and release?

5. In Table III, the Reviewer would suggest to include one column with subspectra assignment. In addition, it would be nice to include the errors associated with determination of the values of the Mössbauer hyperfine parameters.

In summary, the manuscript can be accepted for the publication once revised following the remarks and comments specified above. Thus, a revision is required at this stage.

Reviewer #3:

Remarks to the Author:

Manuscript "epsilon - Fe₂O₃ under compression: Formation of a new iron oxide polymorph" by J.A. Sans et al is about experimental observation and modeling of phase transitions in nanoparticles of

Fe₂O₃ under pressure. Sophisticated experimental techniques like EXAFS and synchrotron-based Mossbauer spectroscopy were used for characterization of the samples.

Here are main comments.

- i) High pressure part is missing in the manuscript. How pressure in the nanoparticles was obtained and most importantly, how the pressure was controlled.
- ii) Crystallographic structure of 17nm nanoparticles might be quite different at the surface and in the core. Not clear which one authors took for explanation of the phase transition.
- iii) What was atomistic model of the nanoparticles for DFT simulations? Number of atoms in the unit cell is missing in description of theoretical method. As a reader I may speculate that the calculations were made for the bulk of Fe₂O₃ crystal. To what extent the calculations of the bulk describe the phase transition in nanoparticle?
- iv) The phase transition in the bulk of Fe₂O₃ found in DFT calculations is interesting by itself. It remains to understand effect of magnetic ordering and temperature on the predicted transition. Atomic magnetic moment fluctuations and anharmonic effect might change the explanation of the phase transition presented in the manuscript.

Firstly, we want to thank the three referees for their comments addressed to increase the quality and clear the doubts up. We are confident that we could solve all their concerns so that they consider this work suitable for publication in Nature Communications.

Reviewer #1

Comment #1. The manuscript in review by Sans et al. focuses on studying compression behavior of less-common, orthorhombic phase in the Fe₂O₃ system known as epsilon-Fe₂O₃. The existence of the epsilon phase and details of its crystal structure have been known for some time. The epsilon phase has been attracting significant attention because of its interesting magnetic properties, including giant coercive field, magnetoresistance, and millimeter wave ferromagnetic resonance, and is an object of current active investigation in the solid state materials field. To my knowledge and understanding, epsilon Fe₂O₃ is not a phase of significant geologic or geophysical relevance.

Authors' reply #1. We agree with the referee that the crystal structure of the epsilon-phase is known for some time but we also highlight that there is a lack of studies of this material under pressure. The total absence of this kind of studies has led to neglect his presence in geologic or geophysical models. The magnetic properties of this material are so strong that a small amount of the mineral in the Earth's strati could change some previously assumed conceptions and shake-up paleomagnetism and magnetotellurism.

Comment #2. The manuscript describes results of three separate in situ high-pressure experiments at ambient temperature with epsilon Fe₂O₃ samples: powder X-ray diffraction, Fe ka-edge EXAFS and synchrotron Mössbauer, over a range of pressure from atmospheric up to about 40 GPa. The experiments seem well designed and executed, and the results are sound and consistent, between the different experiment techniques.

Authors' reply #2. We thank the referee for this encouraging comment and we would like to add the results obtained by ab-initio theoretical simulations as a key aspect in the interpretation of the experimental results. Both theoretical and experimental results under pressure showed a full concordance revealing the paramount importance of the results.

Comment #3. The results indicate continues compressional behavior up to about 25 GPa, followed by a small volume collapse of 2%, which is interpreted in terms of change in magnetic ordering.

For structural refinements the authors use a starting model, which was reported by [22] Kelm and Mader (2005). I believe this model was later debated and a minor revision (in the same space group) was proposed by Gich et al. (2006) Chemistry of Materials, 18, 3889.

Authors' reply #3. The truth is that is quite difficult to identify if the transition is due to structural changes that trigger the magnetic reordering or if the magnetic transition triggers the structural changes. The origin of this kind of transitions has been the subject of a strong controversy in the literature as mentioned in our manuscript for the case of the hematite (alpha-Fe₂O₃).

Regarding the structural model used, the work by Gich et al. was based in powder neutron diffraction combined to synchrotron x-ray diffraction with the aim of studying the magnetic and structural differences in epsilon-Fe₂O₃ above and below a phase transition at 150 K. However, the work by Gich et al. did not report room temperature measurements but only at 10 K and

200 K. The work of Kelm and Mader reports the structure at room temperature as for the measurements of our manuscript, and thus it was more reasonable to take that structure as starting model for our work, both structures being in essence very similar.

Comment #4. I found a report of similar high-pressure experiments on epsilon-Fe₂O₃ (powder XRD and synchrotron Mössbauer), but limited to 12 GPa, published in 2013 DESY annual reports by Brazda et al., but I could not find a corresponding journal publication. Perhaps discussing these results to some extent would be useful (there was an anomaly of compressional behavior at lower pressure, which is not seen in the experiments by Sans et al.).

Authors' reply #4. We really appreciate the mention of this work that we were not aware of. We could not find any corresponding journal publication either but we consider that the work deserves to be cited in our manuscript (Page 4, Results). However, the lack of description of the experimental conditions in the mentioned report makes us impossible to analyse and discuss which is the origin of the anomaly in the compression at lower pressure. Also, this behaviour at low pressures was neither observed in our experiments nor by our theoretical calculations.

Comment #5. In my opinion this is sound work, and worthwhile piece of knowledge revealed, but I do not see enough significant broad appeal to justify publication in Nature Communications in the current form. Any kind of more specialized journal would be better suited. I think the authors made a mistake in overplaying the geological context. This phase is of fairly significant interest in solid state materials science, and I would strongly recommend to focus on these aspects. Otherwise the paper is very clean and well-written.

The geological implications are minor at best. These phases (epsilon and epsilon') are not expected to be present in mantle rocks at appreciable quantity. They are almost for certain only thermodynamically metastable. The study is constrained to ambient temperature, which is not the temperature at which rocks exist in the mantle.

In summary, I can't recommend publication in Nature Communications, but would recommend to refocus the paper on materials science aspects and resubmit to a specialized journal.

Authors' reply #5. We thank the referee for his/her encouraging and motivating words. However, we disagree with his/her statement. We consider that the own rejoinder has been described by the referee: The importance of this material in solid state Physics is huge due to its paramount magnetic properties together with its stability pressure range, which raises the possibility to find it in the Earth's strati. Regarding the fact that we measured at ambient temperature, we can say that the epsilon-Fe₂O₃ is the high temperature phase of the maghemite (above 700 °C) and this structure is stable up to 1100 °C (J. L. García Muñoz et al. Chem. Mater. 29, 9705-9713 (2017)) or being synthesized at temperatures like 1400 °C (Tronc et al. J. of Solid State Chem. 139, 93—104 (1998)). This temperature range coincides which that expected in the Earth's crust and mantle. Moreover, in order to complement our work, we have performed theoretical calculations of the dynamical stability of this compound at different pressures and temperatures showing that the predicted stability of this phase under pressure and temperature changes is compatible with its presence in the Earth interior. All these comments, highlighting the stability of this phase at high temperature, have been included in the main manuscript. The new paragraph (page 4, Introduction) reads: "Moreover, ε-Fe₂O₃ has showed to be highly stable with the application of temperature, being the HT phase of the maghemite²⁸ above 700 °C and remaining stable up to 1100 °C or being synthesized²⁵ at high temperatures as 1400 °C."

We have also included two new plots: a) the predicted Energy vs volume curve at several temperatures (Figure S7) and b) the study of the theoretically simulated dynamical stability of ϵ -Fe₂O₃ at high pressure and high temperature (Figure S8) in the supporting information. The analysis of these figures is added in the following sentences: i) "In order to explore its existence under these conditions, we must include the temperature parameter in the theoretical calculations. The structure becomes more energetically favored with the increase of the temperature, comparing the theoretical evolution of the Gibbs free energy vs volume curve at several temperatures (**Figure S7**). The theoretical simulation of the phonon dispersion curve at 25 GPa and 1800K (**Figure S8**) reveals that the structure is dynamically stable at these conditions by the absence of imaginary or soft modes." in pages 14-15 belonging to the discussions section and ii) " ϵ -Fe₂O₃ has shown an experimental structural stability up to 27 GPa at ambient temperature and a predicted dynamical stability at 25 GPa and 1800K revealing its potential presence in the Earth's interior." in page 15 as part of the summary. Consequently, we include in the manuscript the description of the method used for the inclusion of temperature in the ab initio theoretical calculations (Page 20-21, methods).

Reviewer #2

Comment #1. In this work, the authors report formation of a new iron(III) oxide polymorph from epsilon-Fe₂O₃ upon exposing it to an elevated pressure. Above a threshold value of the pressure, the structural transformation occurs involving all four cation sites in the crystal structure of epsilon-Fe₂O₃. More importantly, the initially tetrahedral site dramatically changes its environment upon pressure increase adopting features of octahedral coordination. The experimental results were supported by the theoretical calculations. The authors show the significance of these results achieved; this study can generate and stimulate an eminent interest in the field of paleomagnetism and magnetotellurism.

In general, the work is well organized and well written; all the experimental data acquired are discussed in the logical and sophisticated way and are well supported by the results of the theory used. The Reviewer finds this work very important for identifying new iron oxide phases that can be stable under the conditions in the Earth's crust and that can be a genesis to other iron oxide phases and/or solid solutions. When going through the manuscript, the Reviewer found some minor things and issues that should be cleared up when revising the work:

Authors' reply #1. We thank the referee for the encouraging comment. We are glad that the reviewer agrees with our point of view regarding the relevance of the manuscript and its conclusions.

Comment #2. The authors explain the changes in the values of the quadrupole splitting upon pressure increase by changes in the direction of the magnetic moment at the specific sites. It is generally accepted that at room temperature, epsilon-Fe₂O₃ behaves as a collinear ferrimagnet and a model of two-supersublattice ferrimagnet is used to explain its magnetic features at room temperature. In this model, A- and C-sublattice magnetizations from distorted octahedral FeA and regular octahedral FeC sites are parallel to each other and form one superlattice and B- and D-sublattice magnetizations from distorted octahedral FeB and tetrahedral FeD sites form another supersublattice. From the Mössbauer data, it can be inferred that this model is probably not valid for this new phase and the sublattice magnetizations may adopt different orientation with respect to the crystallographic axes of epsilon-Fe₂O₃. It is possible, at this stage, to speculate about the orientation of sublattice magnetizations at elevated pressures for individual sites in the crystal structure of epsilon-Fe₂O₃?

Authors' reply #2. The referee is right, we can obtain the magnetization of the sublattice thanks to our theoretical simulations, where a change in the orientation of this magnetization was observed. However, to find out the magnetic structure above 27 GPa one should perform neutron diffraction above this pressure which is technically very demanding and not feasible in most of the neutron facilities worldwide.

Comment #3. Two possible explanations have been suggested for appearance of a doublet upon raising the pressure. The proposed idea of a transition involving emergence of new electronic and magnetic state seems relevant to the Reviewer. However, as the sextets are still present, it implies that only a portion of the epsilon-Fe₂O₃ particles undergo such transition. As the system shows the size distribution (as demonstrated in Figure S1), can this transition be associated with some critical size which is pressure dependent? In other words, as the pressure increases, more and more particles undergo such transition. This could explain the decrease in the spectral areas of sextets and increase in the spectral area of the doublet. Alternatively, this transition can

happen at the surface layers and then proceed to the core of the nanoparticles as the pressure gradually increases. Can the authors comment more on this issue?

Authors' reply #3. We really appreciate this very interesting comment. It is true that the remaining presence of the sextet at high pressure may be due to a size dependence of the phase transition. However, the ADXRD patterns does not exhibit a broadening of the Bragg reflections, which means that the transition is undergone at a pretty similar pressure. On the other hand, we consider that the most plausible explanation to this fact is the effect of non-hydrostatic conditions along the measurement, which triggers the gradual magnetic transition. Whereas in the ADXRD experiment we need very small amount of sample to reach a good signal/noise ratio, the Mössbauer spectroscopy required the use of a large quantity of sample that implies the chances to observe bridging or other non-hydrostatic behaviour. Consequently, it is possible that we were inducing the phase transition in a part of the sample at lower pressures than expected.

Comment #4. Figure 5a: The fitting seems to be not perfect as the fitted middle and inner lines of some spectral components do not match the experimental data. Is it due to the experimental error or the error of the fitting?

Authors' reply #4. We thank the reviewer for highlighting this point and we agree that the fitting of the 1bar spectrum was not perfect. We revisited the model applying a correction for the distribution of the magnetic field, which was not done before to keep the model as simple as possible. The “new” fitting, which exhibits almost identical hyperfine parameters to those reported in the literature but take into account for the distribution of the magnetic field is presented (both here and in the manuscript). On the other hand, we believe that the fitting of the spectra at higher pressures represent well the system, given its complexity.

Comment #5. If the pressure is released, is the crystal structure of epsilon-Fe₂O₃ fully restored? Are there any deviations in the crystal structure due to pressure treatment and release?

Authors' reply #5. Unfortunately, we could not collect experimental data on the downstroke and consequently, we cannot affirm if the crystal structure is fully restored but in our opinion, is the most likely behaviour due the slight structural change originated.

Comment #6. In Table III, the Reviewer would suggest to include one column with subspectra assignment. In addition, it would be nice to include the errors associated with determination of the values of the Mössbauer hyperfine parameters.

Authors' reply #6. We agree with the referee for the sake of clarity and readability. We included a column with the subspectra assignment. The uncertainties associated to the parameters extracted by the fit are already displayed in the table by the numbers in brackets. In order to avoid misunderstandings, this assignment was explicitly included in the table caption.

Comment #7. In summary, the manuscript can be accepted for the publication once revised following the remarks and comments specified above. Thus, a revision is required at this stage.

Authors' reply #7. We thank the referee for the very interesting comments devoted to improve the quality of the paper and we hope that our replies and the parts revised in our manuscript will be enough to consider the publication of this work.

Comment #1. Manuscript “epsilon - Fe₂O₃ under compression: Formation of a new iron oxide polymorph” by J.A. Sans et al is about experimental observation and modeling of phase transitions in nanoparticles of Fe₂O₃ under pressure. Sophisticated experimental techniques like EXAFS and synchrotron-based Mossbauer spectroscopy were used for characterization of the samples. Here are main comments.

Authors’ reply #1. We thank the referee to take the time to review our work and the stimulating comments devoted to clarify some issues.

Comment #2. High pressure part is missing in the manuscript. How pressure in the nanoparticles was obtained and most importantly, how the pressure was controlled.

Authors’ reply #2. We thank the reviewer for this pertinent comment. We have included the following description in the Methods section of the manuscript: “In all the experimental procedures, we use a membrane-type diamond anvil cell consisting in two diamonds faced one to each other. Between them, we use a gasket with a hole in the middle, which defines the pressure cavity where the sample, the pressure calibrant and the pressure transmitting medium are placed. The pressure transmitting medium (in our case helium in the three experiments) assures the application of the pressure occurs hydrostatically. The pressure inside the cavity is obtained by the measurement of a calibrant, in most of the experiments we took advantage of the evolution under pressure of the fluorescence signal of the ruby chips inserted in the pressure cavity. The pressure is controlled by the membrane of the diamond anvil cell which is inflated applying force to the mobile anvil whereas the other diamond is fixed and monitored by the pressure calibrant.”

Comment #3. Crystallographic structure of 17nm nanoparticles might be quite different at the surface and in the core. Not clear which one authors took for explanation of the phase transition.

Authors’ reply #3. The structure revealed by X-ray diffraction is that of the core, for which the periodicity of the crystal exists, and the thus changes observed along the transition (unit cell volume, lattice parameters, ...) refer to the particle core. The surface atoms contribute along with the core to the EXAFS and Mössbauer spectroscopies but for 17 nm particles one can estimate that the percentage of surface atoms P_s is below 10 % and thus its effect is small even if the atomic structure of the surface might be different.

Comment #4. What was atomistic model of the nanoparticles for DFT simulations? Number of atoms in the unit cell is missing in description of theoretical method. As a reader I may speculate that the calculations were made for the bulk of F₂O₃ crystal. To what extent the calculations of the bulk describe the phase transition in nanoparticle?

Authors’ reply #4. The calculations were made considering a bulk Fe₂O₃ crystal being the number of atoms in the unit cell 40. This information has been included in the main manuscript. The nanocrystalline Fe₂O₃ sample has an average grain size of 17 nm. Normally, taking into account our previous experience with other nanomaterials with grain size bigger than 15 nm, the sample can be considered bulk in the calculations. Moreover, our calculations reproduce very well all physical properties at ambient conditions and under pressure observed in our study and in the literature.

Comment #5. The phase transition in the bulk of F2O3 found in DFT calculations is interesting by itself. It remains to understand effect of magnetic ordering and temperature on the predicted transition. Atomic magnetic moment fluctuations and anharmonic effect might change the explanation of the phase transition presented in the manuscript.

Authors' reply #5. We agree with the reviewer about the importance of understanding the effect of the temperature in the compound. For this reason, we have carried out a study including pressure and temperature in our calculations within the quasi-harmonic approximation (QHA) by using Phonopy software (<http://dx.doi.org/10.1016/j.scriptamat.2015.07.021>). These results, shortly summarized below, have been integrated in the manuscript (Pages 15-16, Discussions) with the following sentences: "In order to explore its existence under these conditions, we must include the temperature parameter in the theoretical calculations. The structure becomes more energetically favored with the increase of the temperature, comparing the theoretical evolution of the Gibbs free energy vs volume curve at several temperatures (**Figure S7**). The theoretical simulation of the phonon dispersion curve at 25 GPa and 1800K (**Figure S8**) reveals that the structure is dynamically stable at these conditions by the absence of imaginary or soft modes." and " ϵ -Fe₂O₃ has shown an experimental structural stability up to 27 GPa at ambient temperature and a predicted dynamical stability at 25 GPa and 1800K revealing its potential presence in the Earth's interior."

In the following figure, the Gibbs energy-volume curve at different temperatures is represented. With these calculations, we wanted to show the possible differences in the free energy at 0 K, room temperature 300 K and at the temperature of Earth's crust around 1800 K. As it can be seen in the figure, the volume collapse occurs for unit cell volumes below 380 Å³ for the studied temperatures. Moreover, the pressure evolution of the free energy is the same. The only significant difference is that we obtain a lower free energy at 1800 K.

Dispersion curve theoretically calculated for epsilon-Fe2O3 at 25 GPa and 1800K

We can say that in this compound, the anharmonic effect is not relevant since within the quasi-harmonic approximation, we observe that this phase is dynamically stable in the whole range of explored pressures and temperatures. We have calculated the phonon dispersion at ambient conditions as well as the phonon dispersion at 25 GPa and 1800 K (Figure S8), conditions proper to earth's crust, and any phonon instability is observed. Therefore, the anharmonic effect would not change the explanation of such phase transition.

Concerning the magnetic moment fluctuations, the calculations were performed including magnetic anisotropy and non collinear magnetization, allowing to relax the magnetic moments during the self-consistent calculations in all pressure range. Therefore, we consider that the strong interaction between the volume collapse and the local structural changes affected by the magnetic properties is very well developed.

Reviewers' Comments:

Reviewer #1:

Remarks to the Author:

I have now completed the review of revised version of the manuscript by Sans et al.

My general opinion about the paper has not changed. It is a very nice piece of work, thoroughly done, with solid experimental results, supplemented by calculations, and what seems like a very reasonable interpretation. I genuinely enjoyed reading the paper. The paper definitely deserves publication, but I still think this is not a topic for Nature Comm. After the revisions the paper is more detailed about the physical and crystallographic interpretation of the observed phenomenon, and it is definitely with a benefit for the reader. To me, the manuscript looks more like an American Mineralogist paper.

My arguments why this does not see to me to be a "major discovery" with "significant impact" have not changed:

1. The relevance of the reported phases transition is discussed with primary emphasis on geology and geophysics BUT:

2. Epsilon-Fe₂O₃ is not a proper mineral (not found in natural rocks).

3. Epsilon-Fe₂O₃ is almost certainly not an equilibrium phase, even in pure Fe₂O₃ system.

4. Even pure Fe₂O₃ is unlikely to be present in the deep Earth interior in significant enough quantity to excite geophysicists.

I do see the benefit in adding high-T calculations to the study, and demonstrating dynamic stability, but it does not make either of the investigated phases an equilibrium phase (they are metastable polymorphs).

A lot of papers reporting discovery of new metastable polymorphic transitions in minerals that are truly abundant and unquestionably relevant to the earth interior (e.g. SiO₂, MgSiO₃, Mg₂SiO₄) are being published in specialized journals like American Mineralogist, and I think this is appropriate, unless there are far reaching and significant consequences of these new phenomena.

I am sorry for not to be more positive about appropriateness of this work the Nature Comm.

Reviewer #2:

Remarks to the Author:

The authors revised the manuscript very thoroughly addressing all the issues/comments and suggestions raised during the first critical assessment of the manuscript. The Reviewer is highly satisfied with the revision and strongly believes that the present work will stimulate future studies in this field, in particular, encouraging attempts for searching for new iron oxide based phases and understanding their genesis in nature (and Earth's crust). Thus, the manuscript in the current revised form is acceptable for publication in Nature Communications. The Reviewer recommends the publication of the work as it is.

Reviewer #3:

Remarks to the Author:

Modified version of the manuscript is appropriate for Nature Communications. There are still some minor corrections.

i) There is still no number of atoms mentioned in the description of DFT calculations in the merged final version of the manuscript. The number is important for readers who are interested in size effects and how adequate are calculations of the bulk structure transitions to the core structure of 17 nm clusters. 100 nm clusters definitely has bulk structure, 1 nm clusters are not and should take into account surface effects. 17 nm is somewhere in between and case dependent. It seems that authors provided evidence that for Fe₂O₃ bulk calculations are relevant for the cluster sizes studied in the paper.

ii) For a small unit cell of 40 atoms the bulk the ab-initio molecular dynamics (AIMD) calculations

of EOS in NVE/NPT ensembles are feasible. QHA is less sophisticated approximation, but better than nothing. Formulas for Helmholtz and Gibbs energies (1) and (2) are well known, does not contain any new information and should not use the precious space of Nature journal.

Reviewer #1

Comment #1. I have now completed the review of revised version of the manuscript by Sans et al.

My general opinion about the paper has not changed. It is a very nice piece of work, thoroughly done, with solid experimental results, supplemented by calculations, and what seems like a very reasonable interpretation. I genuinely enjoyed reading the paper. The paper definitely deserves publication, but I still think this is not a topic for Nature Comm. After the revisions the paper is more detailed about the physical and crystallographic interpretation of the observed phenomenon, and it is definitely with a benefit for the reader. To me, the manuscript looks more like an American Mineralogist paper.

My arguments why this does not see to me to be a “major discovery” with “significant impact” have not changed:

Authors’ reply #1. First of all, we would like to thank the reviewer for the positive comments and the time devoted to make a critical reading of the manuscript. Nevertheless, we disagree with his/her conclusions.

Comment #2. The relevance of the reported phases transition is discussed with primary emphasis on geology and geophysics BUT:

Authors’ reply #2. We have corrected the strong accent in the geophysical field and we have highlighted the implications in Solid State Physics of this study.

Comment #3. Epsilon-Fe₂O₃ is not a proper mineral (not found in natural rocks).

Authors’ reply #3. We strongly disagree with this comment. The epsilon-Fe₂O₃ has been found as a mineral called Luogufengite, Al-bearing ε-Fe₂O₃, which was discovered in late Pleistocene basaltic scoria from the Menan Volcanic Complex nearby Rexburg, Idaho (Xu et al. American Mineralogist (2017) 102 (4): 711-719) but also recently, it was discovered the presence of pure nanomineral of ε-Fe₂O₃ in basaltic rocks (S. Lee et al. Minerals 8, 97 (2018))

Comment #4. Epsilon-Fe₂O₃ is almost certainly not an equilibrium phase, even in pure Fe₂O₃ system.

Authors’ reply #4. Epsilon-Fe₂O₃ is a metastable phase which does not imply that could not appear in the nature or forming part of the Earth’s interior. The strong pressures and temperatures of the inner regions of Earth can develop the ideal environment to naturally synthesize this compound.

Comment #5. Even pure Fe₂O₃ is unlikely to be present in the deep Earth interior in significant enough quantity to excite geophysicists.

Authors' reply #5. Here, we disagree again with the referee. The study of pure Fe₂O₃ has been extensively done by geophysicists. For instance, Shim, S.-H. et al. Electronic and magnetic structures of the postperovskite-type Fe₂O₃ and implications for planetary magnetic records and deep interiors. Proc. Natl Acad. Sci. USA 106, 5508–5512 (2009) or Dobson, D. P. & Brodholt, J. P. Subducted banded iron formations as a source of ultralow-velocity zones at the core-mantle boundary. Nature 434, 371–374 (2005). These works are only a small sample of the vast amount of works made by geophysicists in the literature.

Comment #6. I do see the benefit in adding high-T calculations to the study, and demonstrating dynamic stability, but it does not make either of the investigated phases an equilibrium phase (they are metastable polymorphs).

Authors' reply #6. The recent discovery of this phase in basaltic rocks indicates that this metastable phase can be found in nature and then the studies of their dynamical stability becomes relevant in our opinion.

Comment #7. A lot of papers reporting discovery of new metastable polymorphic transitions in minerals that are truly abundant and unquestionably relevant to the earth interior (e.g. SiO₂, MgSiO₃, Mg₂SiO₄) are being published in specialized journals like American Mineralogist, and I think this is appropriate, unless there are far reaching and significant consequences of these new phenomena.

Authors' reply #7. Just the presence of this phase in nature has been recently published in American Mineralogist. The study of the properties of this phase under extreme conditions and the analysis of the origin of the volume collapse, when it is the first reference to a HS-IS transition in Fe₂O₃ compounds is a significant result to be published in a high impact journal. Moreover, from 2016, 13 articles have been published in Nature journals (excluding Scientific Reports) related to the properties of the iron oxide compounds and one can find studies of iron oxide phases at extreme conditions such as E. Bykova et al. Nature Communications 7, 10661 (2016).

Comment #8. I am sorry for not to be more positive about appropriateness of this work the Nature Comm.

Authors' reply #8. We would like to point out that despite we not agreeing with the referee's view on the relevance of our work and its suitability to be published in Nature Communications, we acknowledge that so far there are no evidences that ϵ -Fe₂O₃ is very abundant in the earth interior. Accordingly, we have made changes in the manuscript to downplay the geophysical its relevance of ϵ -Fe₂O₃ emphasizing more the Solid State Physics orientation.

Reviewer #2

Comment #1. The authors revised the manuscript very thoroughly addressing all the issues/comments and suggestions raised during the first critical assessment of the manuscript. The Reviewer is highly satisfied with the revision and strongly believes that the present work will stimulate future studies in this field, in particular, encouraging attempts for searching for new iron oxide based phases and understanding their genesis in nature (and Earth's crust). Thus, the manuscript in the current revised form is acceptable for publication in Nature Communications. The Reviewer recommends the publication of the work as it is.

Authors' reply #1. We really appreciate the encouraging words and we want to thank the nice comments and suggestions that the referee made in the previous revision that improved the quality of our work.

Reviewer #3

Comment #1. Modified version of the manuscript is appropriate for Nature Communications. There are still some minor corrections.

i) There is still no number of atoms mentioned in the description of DFT calculations in the merged final version of the manuscript. The number is important for readers who are interested in size effects and how adequate are calculations of the bulk structure transitions to the core structure of 17 nm clusters. 100 nm clusters definitely has bulk structure, 1 nm clusters are not and should take into account surface effects. 17 nm is somewhere in between and case dependent. It seems that authors provided evidence that for Fe₂O₃ bulk calculations are relevant for the cluster sizes studied in the paper.

ii) For a small unit cell of 40 atoms the bulk the ab-initio molecular dynamics (AIMD) calculations of EOS in NVE/NPT ensembles are feasible. QHA is less sophisticated approximation, but better than nothing. Formulas for Helmholtz and Gibbs energies (1) and (2) are well known, does not contain any new information and should not use the precious space of Nature journal.

Authors' reply #1. Thank you very much for your comments. We have added the number of atoms of the unit cell in the description of DFT calculations and we have removed the formulas for Helmholtz and Gibbs energies.